# Accurate projective two-band description of topological superfluidity in spin-orbit-coupled Fermi gases

**Joachim Brand[1], Lauri Toikka[1,2], Ulrich Zülicke[3]⋆**

**1** Dodd-Walls Centre for Photonic and Quantum Technologies, Centre for Theoretical Chemistry and Physics, and New Zealand Institute for Advanced Study, Massey University, Private Bag 102904 NSMC, Auckland 0745, New Zealand
**2** Center for Theoretical Physics of Complex Systems, Institute for Basic Science (IBS), Daejeon 34051, Republic of Korea
**3** Dodd-Walls Centre for Photonic and Quantum Technologies, School of Chemical and Physical Sciences, Victoria University of Wellington, Wellington 6140, New Zealand

⋆ uli.zuelicke@vuw.ac.nz

## Abstract

The interplay of spin-orbit coupling and Zeeman splitting in ultracold Fermi gases gives rise to a topological superfluid phase in two spatial dimensions that can host exotic Majorana excitations. Theoretical models have so far been based on a four-band Bogoliubov-de Gennes formalism for the combined spin-1/2 and particle-hole degrees of freedom. Here we present a simpler, yet accurate, two-band description based on a well-controlled projection technique that provides a new platform for exploring analogies with chiral *p*-wave superfluidity and detailed future studies of spatially non-uniform situations.



# 1   Introduction

Topological superfluids and superconductors [1, 2] are the focus of great current interest because of their ability to host unconventional Majorana excitations [3]. An attractive route towards realization that was suggested early on [4–8] utilizes two-dimensional (2D) *s*-wave superfluids with spin-orbit coupling. The transition from the non-topological superfluid phase to the topological superfluid phase in these systems is driven by increasing the Zeeman energy splitting between spin-↑ and spin-↓ single-particle states beyond the critical value where the excitation gap for spin-↓ particles closes. The resulting effectively spinless superfluid state is expected to show all the hallmarks associated with chiral *p*-wave pairing [9], including exotic Majorana states in vortex cores [10–12]. Promising experimental efforts are currently undertaken in condensed-matter systems [13–15] and ultracold-atom gases [16, 17], which have the potential to provide complementary insight and crucial ingredients for topological quantum information devices [18]. One of the important technical differences between these two platforms is the way how the Zeeman spin splitting is introduced. For superconductors, the required magnetic-field strengths are typically deleterious to superconductivity, motivating a search for alternative approaches [19–21]. Being unencumbered by this drawback, ultracold-atom realizations may offer a more direct avenue towards implementation of topological superfluidity. Our present study is intended to provide a useful tool for investigating topological effects in superfluid spin-orbit-coupled Fermi gases and ultimately enable the design and optimization of proof-of-concept devices.

Previous theoretical studies [22–34] of *s*-wave pairing in Fermi gases with spin-orbit coupling and Zeeman splitting have examined the physical properties of these paradigmatic systems using a mean-field treatment in a four-dimensional Nambu space. While the breaking of spin-rotational invariance generally requires such a more complicated [35], and in general only numerically accessible, treatment, the subspace of the spin-↑ degrees of freedom that are relevant for topological properties is only two-dimensional. See Fig. 1(a) for an illustration. Thus it would be desirable to have an effective description based on projecting into the spin-↑ subspace. However, to discuss manipulations of the system by controllable physical parameters and to make predictions for experimentally accessible observables, the influence of the spin-↓ degrees of freedom cannot be ignored. Here we present a fully self-consistent effective theory that is based on an application of the Feshbach-partitioning technique [36, 37] to the spin-resolved Bogoliubov-de Gennes (BdG) Hamiltonian [35, 38] under the assumption that the *s*-wave pair potential $|\Delta|$ is small compared to the Zeeman energy shift $h$ that favors spin-↑ over spin-↓ configurations. Our effective theory is designed to reproduce salient features of the Bogoliubov-quasiparticle spectrum [Fig. 1(b)] and all relevant parametric dependences associated with the topological phase transition. This formalism is therefore ideally suited to be a platform for further studies of topological effects, including those associated with non-uniform superfluid phases [39–43]. In order to be specific, and also because it is the physically most interesting case, we develop the theory for a 2D superfluid, but the formalism lends itself to be easily applied to 1D or 3D situations as well.

One of the main benefits associated with having a spin-↑-projected effective theory is that it facilitates the numerical treatment of inhomogeneous and time-dependent situations. For example, in the time-dependent study of soliton or vortex dynamics (e.g., similar to recent work reported in Ref. [40]), the reduction of numerical complexity obtained by moving from the original four-spinor formalism to the projected two-spinor approach is significant. But already for the homogeneous superfluid, the projected-theory results are very useful because, e.g., they provide simpler expressions for the wave functions of the Bogoliubov quasi-particle excitations and thus facilitate convenient analytical approximations. As an example, we derive simple analytic expressions for the chemical potential of the spin-orbit-coupled two-dimensional su-

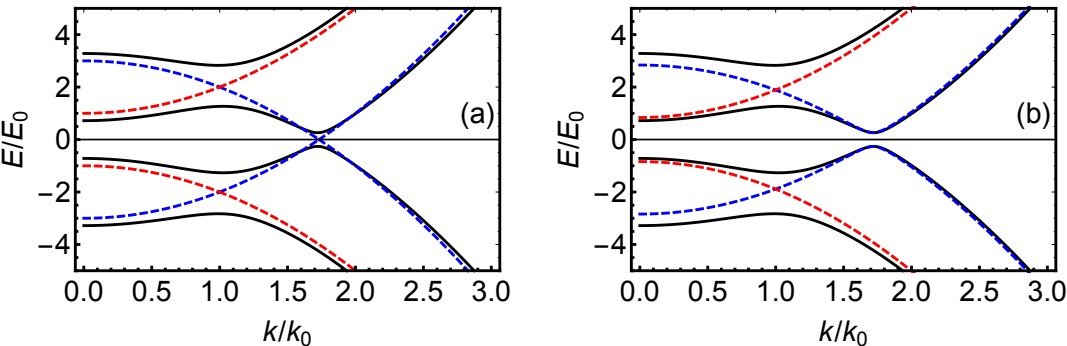

Figure 1: Spectrum of Bogoliubov-quasiparticle excitation energies $E$ as a function of wave-vector magnitude $k \equiv \sqrt{k_x^2 + k_y^2}$ in the uniform topological-superfluid phase of a 2D Fermi gas with spin-orbit coupling $\lambda_{\mathbf{k}} \equiv \lambda(k_x - i\,k_y)$ and Zeeman splitting $h$. In panel (a), the dashed blue (red) curves are the spin-$\uparrow$ (spin-$\downarrow$) dispersions in the absence of $s$-wave pairing ($\Delta = 0$) and spin-orbit coupling ($\lambda = 0$) but with large Zeeman splitting $h = 2E_0$. A finite $\Delta$ couples spin-$\uparrow$ and spin-$\downarrow$ states, resulting in gaps opening at $E = \pm h$ and $k = \sqrt{2m\mu/\hbar^2}$, where $\mu$ denotes the chemical potential. In the situation depicted here, $\mu = E_0$ with $E_0 \equiv \hbar^2 k_0^2/(2m)$ being an arbitrary energy scale. When $\lambda$ is also finite, a third gap opens at $E = 0$, and the system is a topological superfluid for $h > \sqrt{\mu^2 + |\Delta|^2}$. The black solid curves are the exact dispersions for $2m\lambda/(\hbar^2 k_0) = 0.4$ and $|\Delta|/E_0 = 0.8$. Panel (b) again depicts these exact dispersions as black solid curves, together with the approximate dispersions obtained by us using a Feshbach projection onto the spin-$\uparrow$ (spin-$\downarrow$) subspace shown as the dashed blue (red) curves.

perfluid [see Eq. (21) and (22)].

The results presented below can be compared with, and also extend, those of previous studies of superfluidity in Fermi gases with spin-orbit coupling and Zeeman splitting where self-consistency implied fixing the total particle density [23–26,30–34]. (In contrast, Refs. [27–29] consider the situation with fixed chemical potential. See also early work [22] that focused on a lattice realization.) Most relevant benchmarking for our present context is provided by previous works pertaining to uniform 2D systems [32–34], but there are also useful connections to be made with known results for trapped 2D systems [30] and 3D systems with 2D Rashba-type spin-orbit coupling [23–26,31]. In a slight variation on our situation of interest, Ref. [32] considers the population imbalance between spin-$\uparrow$ and spin-$\downarrow$ particles as a control parameter rather than the Zeeman energy.

This article is organised as follows. In the following Section 2, we introduce the microscopic model for superfluid 2D Fermi gases with spin-orbit coupling and Zeeman splitting and apply Feshbach partitioning to derive effective theories describing the spin-$\uparrow$ and spin-$\downarrow$ sectors separately. The obtained formalism is applied in Section 3 to devise a fully self-contained procedure for finding the chemical potential $\mu$ and $s$-wave pair potential $\Delta$ for uniform systems at fixed total particle density $n \equiv n_\uparrow + n_\downarrow$ entirely within the $2 \times 2$-projected theory for the spin-$\uparrow$ sector. The efficacy of this approach is demonstrated in Section 4 by presenting a comparison of predictions for phase boundaries and thermodynamic quantities obtained within the effective two-band and exact four-band theories. Following the usual convention, we measure relevant parameters in units of the density-defined magnitude of the 2D Fermi wave vector $k_F = \sqrt{2\pi n}$ and associated Fermi energy $E_F = \hbar^2 k_F^2/(2m) \equiv \pi\hbar^2 n/m$, with $m$ denoting the single-particle mass. Our conclusions and an outlook toward future work are presented in the final Section 5.

## 2 Feshbach partitioning of the BdG Hamiltonian

Our starting point is the Bogoliubov-de Gennes (BdG) equation describing quasiparticle excitations in a superfluid Fermi gas without spin-rotational invariance. It reads [35, 38]

$$
\mathcal{H}
\begin{pmatrix}
u^{\uparrow} \\
u^{\downarrow} \\
v^{\uparrow} \\
v^{\downarrow}
\end{pmatrix}
= E
\begin{pmatrix}
u^{\uparrow} \\
u^{\downarrow} \\
v^{\uparrow} \\
v^{\downarrow}
\end{pmatrix},
\tag{1a}
$$

with complex spinor entries $u^{\sigma}$ ($v^{\sigma}$) denoting quantum amplitudes of spin-$\sigma$ particle (hole) states in a Bogoliubov excitation of the superfluid. (Here and in the following, $\sigma \in \{\uparrow, \downarrow\}$ is used as a compact label for the spin degree of freedom.) The Hamiltonian matrix in four-dimensional particle-hole (Nambu) space is

$$
\mathcal{H} =
\begin{pmatrix}
\epsilon_{\mathbf{k}\uparrow} - \mu & \lambda_{\mathbf{k}} & 0 & -\Delta \\
\lambda_{\mathbf{k}}^{*} & \epsilon_{\mathbf{k}\downarrow} - \mu & \Delta & 0 \\
0 & \Delta^{*} & -\epsilon_{\mathbf{k}\uparrow} + \mu & \lambda_{\mathbf{k}}^{*} \\
-\Delta^{*} & 0 & \lambda_{\mathbf{k}} & -\epsilon_{\mathbf{k}\downarrow} + \mu
\end{pmatrix},
\tag{1b}
$$

where '$*$' indicates complex conjugation, $\mathbf{k} \equiv (k_x, k_y)$ is the 2D wave vector, and $\epsilon_{\mathbf{k}\uparrow(\downarrow)} = \epsilon_{\mathbf{k}} \overset{-}{(+)} h$ with $\epsilon_{\mathbf{k}} = \hbar^2 (k_x^2 + k_y^2)/2m$. For spatially inhomogeneous configurations, $k_j \equiv -i\partial_j$ is to be treated as an operator while, for a homogeneous superfluid, it can be replaced by its wave-number eigenvalue. The Zeeman energy splitting is denoted by $h$, and $\lambda_{\mathbf{k}}$ is the spin-orbit coupling. Examples of typically considered $\mathbf{k}$-linear spin-orbit couplings are the 2D-Dirac [44], 2D-Rashba [45, 46], and 2D-Dresselhaus [46, 47] types that correspond to different functional forms $\lambda_{\mathbf{k}} = \lambda(k_x - ik_y)$, $\lambda i(k_x - ik_y)$, and $\lambda(k_x + ik_y)$, respectively, but are all unitarily equivalent. In particular, the eigenvalue spectrum of $\mathcal{H}$ for the homogeneous superfluid depends on the spin-orbit coupling only via the quantity $|\lambda_{\mathbf{k}}|^2 \equiv \lambda^2(k_x^2 + k_y^2)$ and therefore has the same functional form for all three of the above-mentioned spin-orbit-coupling types. The eigenvalue spectrum $E_{\mathbf{k}\alpha,\eta}$ is characterised by four bands of dispersion relations as shown in Fig. 1(a) for a particular set of parameters. In order to be able to refer to a specific band, we introduce the following naming convention (for the fully gapped case), where $\alpha = +(-)$ indicates states that have energy $E \geq 0$ ($E \leq 0$), whereas the index $\eta => (<)$ labels the higher(lower)-energy pair of excitation branches; i.e., $|E_{\mathbf{k}\alpha,>}| \geq |E_{\mathbf{k}\alpha,<}|$. For what follows, it will be useful to know the asymptotic behavior of the dispersions in the limit of large 2D-wave-vector magnitude $k \equiv \sqrt{k_x^2 + k_y^2}$. We find

$$
E_{\mathbf{k}+,<(>)} = \epsilon_{\mathbf{k}} - \mu \overset{-}{(+)} \sqrt{h^2 + |\lambda_{\mathbf{k}}|^2} + \mathcal{O}\left(\frac{|\Delta|^2}{\epsilon_{\mathbf{k}}}\right).
\tag{2}
$$

The matrix equation (1a) can be reorganized by forming $2 \times 2$ sub-blocks on the diagonal that are associated with subspaces for fixed spin degree of freedom,

$$
\begin{pmatrix}
\mathcal{H}^{\uparrow\uparrow} & \mathcal{H}^{\uparrow\downarrow} \\
\mathcal{H}^{\downarrow\uparrow} & \mathcal{H}^{\downarrow\downarrow}
\end{pmatrix}
\begin{pmatrix}
w^{\uparrow} \\
w^{\downarrow}
\end{pmatrix}
= E
\begin{pmatrix}
w^{\uparrow} \\
w^{\downarrow}
\end{pmatrix},
\tag{3a}
$$

with the definitions

$$w^\sigma = \begin{pmatrix} u^\sigma \\ v^\sigma \end{pmatrix}, \tag{3b}$$

$$\mathscr{H}^{\sigma\sigma} = \begin{pmatrix} \epsilon_{\mathbf{k}\sigma} - \mu & 0 \\ 0 & -\epsilon_{\mathbf{k}\sigma} + \mu \end{pmatrix}, \tag{3c}$$

$$\mathscr{H}^{\uparrow\downarrow} \equiv \left(\mathscr{H}^{\downarrow\uparrow}\right)^\dagger = \begin{pmatrix} \lambda_{\mathbf{k}} & -\Delta \\ \Delta^* & \lambda_{\mathbf{k}}^* \end{pmatrix}, \tag{3d}$$

and '†' denoting Hermitian conjugation. Simple algebra yields $2 \times 2$-matrix equations that formally decouple the individual spin sectors,

$$w^{\bar\sigma} = -\left(\mathscr{H}^{\bar\sigma\bar\sigma} - E\,\mathbb{1}\right)^{-1} \mathscr{H}^{\bar\sigma\sigma} w^\sigma, \tag{4a}$$

$$\left[\mathscr{H}^{\sigma\sigma} - \mathscr{H}^{\sigma\bar\sigma}\left(\mathscr{H}^{\bar\sigma\bar\sigma} - E\,\mathbb{1}\right)^{-1} \mathscr{H}^{\bar\sigma\sigma}\right] w^\sigma = E\,w^\sigma, \tag{4b}$$

where $\mathbb{1}$ is the $2\times2$ identity matrix and $\bar\sigma$ denotes the opposite of $\sigma$; i.e., $\bar\sigma = \downarrow (\uparrow)$ if $\sigma = \uparrow (\downarrow)$. While formally exact and a $2\times2$ BdG-like equation in spin-$\sigma$ space, Eq. (4b) is not really useful without approximations, since the unknown energy eigenvalue $E$ appears on both sides of the equation. Assuming $\Delta$ to be small compared to other relevant energy scales, we replace $E$ in the denominator by the exact $\Delta = 0$ solution from Eq. (2) to obtain the approximation

$$\left(\mathscr{H}^{\downarrow\downarrow} - E\,\mathbb{1}\right)^{-1} \approx \frac{1}{2h_{\mathbf{k}}}\begin{pmatrix} 1 & 0 \\ 0 & -1 \end{pmatrix}, \tag{5a}$$

$$\left(\mathscr{H}^{\uparrow\uparrow} - E\,\mathbb{1}\right)^{-1} \approx \frac{1}{2h_{\mathbf{k}}}\begin{pmatrix} -1 & 0 \\ 0 & 1 \end{pmatrix}. \tag{5b}$$

Here $2h_{\mathbf{k}} = h + \sqrt{h^2 + |\lambda_{\mathbf{k}}|^2}$ can be thought of as a $k$-dependent effective Zeeman splitting that is modified by the presence of the spin-orbit coupling. The approximation becomes good if $2h_{\mathbf{k}}$ is large compared to the neglected term, i.e. when $|\Delta|^2 \ll \epsilon_{\mathbf{k}} h_{\mathbf{k}}$, which is always asymptotically true for large $k$. Substituting the approximations (5) into (4b) yields truly decoupled BdG equations for the individual spin sectors,

$$\mathscr{H}_{\text{eff}}^\sigma w^\sigma = E^\sigma w^\sigma, \tag{6}$$

where

$$\mathscr{H}_{\text{eff}}^\sigma = \begin{pmatrix} \xi_{\mathbf{k}\sigma} & \tilde\Delta_{\mathbf{k}\sigma} \\ \tilde\Delta_{\mathbf{k}\sigma}^* & -\xi_{\mathbf{k}\sigma} \end{pmatrix}, \tag{7a}$$

$$\xi_{\mathbf{k}\uparrow(\downarrow)} = \epsilon_{\mathbf{k}\uparrow(\downarrow)} \overset{+}{(-)} \frac{|\Delta|^2 - |\lambda_{\mathbf{k}}|^2}{2h_{\mathbf{k}}} - \mu, \tag{7b}$$

$$\tilde\Delta_{\mathbf{k}\uparrow(\downarrow)} = -\left\{\lambda_{\mathbf{k}}^{(*)}, \frac{\Delta}{h_{\mathbf{k}}}\right\}. \tag{7c}$$

We adopted an anticommutator notation $\{A, B\} = (AB + BA)/2$ to be able to incorporate situations with spatially inhomogeneous $s$-wave pair potential[1] $\Delta$. Notice that the off-diagonal matrix element in (7a) that is responsible for opening the gap in the quasiparticle spectrum is given by $\lambda_{\mathbf{k}}\Delta/h_{\mathbf{k}}$, which is proportional to both the spin-orbit-coupling strength $\lambda$ and $\Delta$.

---

[1]Spatial inhomogeneity will also require a suitable treatment of the $\mathbf{k}$-dependence in $h_{\mathbf{k}}$.

In the case of **k**-linear spin-orbit coupling and uniform $\Delta$, its leading-order dependence on **k** resembles the pair potential for a chiral-$p$-wave superfluid. The emergence of $p$-wave pairing in the present context was inferred in earlier works [5, 7, 8, 31] through a transformation into the so-called 'helicity' basis of the single-particle Hamiltonian in the presence of spin-orbit coupling. Although instructive at the time, this transformation does not provide a useful basis for in-depth quantitative studies. In contrast, our present work enables a precise derivation of the effective pairing potential of spin-$\sigma$ states, including systematic corrections to the chiral-$p$-wave form.

Assuming a spatially uniform pair potential $\Delta$, straightforward diagonalization of (7a) yields approximate energy dispersions and corresponding eigenspinors for the spin-$\sigma$ subspace as

$$E_{\mathbf{k}\alpha}^{\sigma} = \alpha \sqrt{\xi_{\mathbf{k}\sigma}^2 + \frac{|\lambda_{\mathbf{k}}|^2 |\Delta|^2}{h_{\mathbf{k}}^2}}, \tag{8a}$$

$$w_{\mathbf{k}\alpha}^{\downarrow(\uparrow)} = \sqrt{N_{\mathbf{k}\alpha}^{\downarrow(\uparrow)}} \begin{pmatrix} \sqrt{\dfrac{E_{\mathbf{k}\alpha}^{\downarrow(\uparrow)} + \xi_{\mathbf{k}\downarrow(\uparrow)}}{2E_{\mathbf{k}\alpha}^{\downarrow(\uparrow)}}} \\ -\alpha \dfrac{\lambda_{\mathbf{k}}^{(*)}}{|\lambda_{\mathbf{k}}|} \dfrac{\Delta^*}{|\Delta|} \sqrt{\dfrac{E_{\mathbf{k}\alpha}^{\downarrow(\uparrow)} - \xi_{\mathbf{k}\downarrow(\uparrow)}}{2E_{\mathbf{k}\alpha}^{\downarrow(\uparrow)}}} \end{pmatrix}. \tag{8b}$$

The $N_{\mathbf{k}\alpha}^{\sigma}$ are normalization factors that have to be found from the four-spinor normalization condition $1 = \sum_{\sigma} \left( w^{\sigma} \right)^{\dagger} w^{\sigma}$. Using the substitution (4a), this condition translates into separate normalization conditions for the individual spin sectors,

$$1 = \left( w_{\mathbf{k}\alpha}^{\sigma} \right)^{\dagger} \mathscr{L}_{\mathbf{k}\alpha}^{\sigma} w_{\mathbf{k}\alpha}^{\sigma}, \tag{9a}$$

$$\mathscr{L}_{\mathbf{k}\alpha}^{\sigma} = \mathbb{1} + \mathscr{H}^{\sigma\bar{\sigma}} \left( \mathscr{H}^{\bar{\sigma}\bar{\sigma}} - E_{\mathbf{k}\alpha}^{\sigma} \mathbb{1} \right)^{-2} \mathscr{H}^{\bar{\sigma}\sigma}. \tag{9b}$$

Further application of the approximations (5) in (9b) yields

$$N_{\mathbf{k}\alpha}^{\sigma} \approx \left[ 1 + \frac{|\Delta|^2 + |\lambda_{\mathbf{k}}|^2}{4h_{\mathbf{k}}^2} \right]^{-1}. \tag{10}$$

Figure 1(b) shows a comparison between the exact dispersions $E_{\mathbf{k}\alpha,\eta}$, obtained by diagonalizing the original $4 \times 4$ BdG Hamiltonian (1b), and the approximate results $E_{\mathbf{k}\alpha}^{\sigma}$ from (8a), calculated using the $2 \times 2$-subspace projections. While the projected theory does not reproduce the $s$-wave pairing gaps around $E = \pm h$ and $k = \sqrt{2m\mu/\hbar^2}$, it describes very well the region around the topological gap at $E = 0$. Most crucially, as it turns out, the dispersions for large $k$ are correctly given by the effective $2 \times 2$-projected approach. As shown in the following, this enables a faithful description of relevant thermodynamic properties. Further comparisons between the exact $4 \times 4$-theory dispersions and the $2 \times 2$-projection approximations are explored in Fig. 2. The low-energy gap structure turns out to be well-described even in the non-topological regime, as illustrated in Fig. 2(a). Overall good agreement is achieved in situations when the chemical is negative[2] due to the absence of any crossing points between opposite-spin dispersions [Fig. 2(b)]. The topological gap ceases to be well-described for quite large spin-orbit-coupling strengths [Fig. 2(c)].

Our Feshbach-projection approach embodied in Eqs. (4) and (5) differs from common perturbative methods such as the Schrieffer-Wolf transformation[3] in two crucial aspects. Firstly,

---

[2]Instances where the chemical potential of a superfluid becomes negative include the BEC regime of the BCS-BEC crossover [48, 49], and systems with large spin-orbit coupling [50].

[3]See, e.g., Appendix B in Ref. [46] for a detailed discussion, or the original work [51].

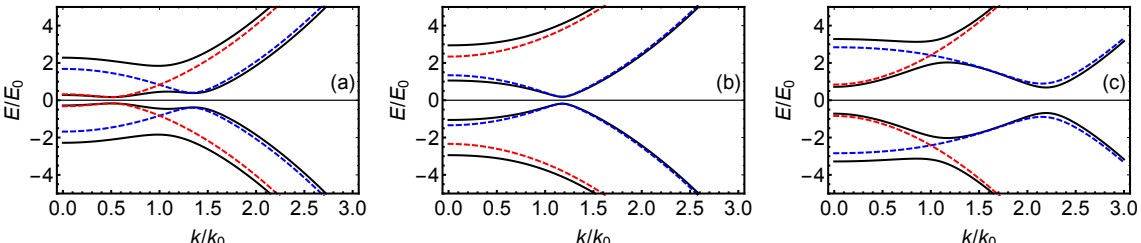

Figure 2: More comparisons between the exact Bogoliubov-quasiparticle dispersions for the $4 \times 4$ BdG Hamiltonian (1b) (black solid curves) with the approximate $2 \times 2$-projection results $E^{\sigma}_{\mathbf{k}\alpha}$ from (8a) [dashed blue (red) curve for $\sigma = \uparrow (\downarrow)$]. Panel (a) shows an example for the situation where the superfluid is non-topological [$h = E_0$, $2m\lambda/(\hbar^2 k_0) = 0.4$, $\mu = E_0$, and $|\Delta|/E_0 = 0.8$]. The case depicted in Panel (b) is for a topological superfluid having a negative chemical potential, which typically occurs for large two-particle binding energies and/or large spin-orbit-coupling strengths [here $h = 2E_0$, $2m\lambda/(\hbar^2 k_0) = 0.4$, $\mu = -0.5 E_0$, and $|\Delta|/E_0 = 0.8$]. Panel (c) illustrates deviations occurring when spin-orbit coupling becomes quite large [$h = 2E_0$, $2m\lambda/(\hbar^2 k_0) = 1.5$, $\mu = E_0$, and $|\Delta|/E_0 = 0.8$].

the dispersions (8a) derived from our $2 \times 2$-projected Bogoliubov-de Gennes Hamiltonians (7a) are well-behaved at all $\mathbf{k}$, whereas those obtained, e.g., from the Schrieffer-Wolf transformation become singular at the $s$-wave pairing gap because of a degeneracy between the eigenvalues of $\mathscr{H}^{\uparrow\uparrow}$ and $\mathscr{H}^{\downarrow\downarrow}$ at this point. Secondly, by careful choice of the approximation (5), we are able to reproduce the large-$|\mathbf{k}|$ asymptotics of the exact energy dispersions within the $2 \times 2$-projected approach, which cannot be achieved by perturbation theory because it treats the spin-sector couplings given in Eq. (3d) as small quantities.

By construction, the projected-theory results for quasiparticle dispersions and wave functions become strictly exact in the limit of vanishing $s$-wave pair potential $\Delta$, i.e., when no avoided crossings occur. Thus, for finite $\Delta$, we may expect all quantities that do not explicitly depend on the avoided crossing between the spin-$\uparrow$ and spin-$\downarrow$ dispersions to be described correctly as long as $|\Delta| \ll h$.

# 3 Self-consistency for uniform systems with fixed density

Knowledge of the Bogoliubov-quasiparticle excitations permits calculation of all physical quantities of interest [35, 38]. For simplicity, we focus on the zero-temperature limit in the following. Generalization to the case of finite temperatures is straightforward [35, 38] but does not add any crucial insights for our present purpose.

The pair potential $\Delta$ can be expressed in terms of the eigenspinors of the BdG equation (1a) and the strength $g$ of attractive interactions in the $s$-wave channel as

$$\Delta = -\frac{g}{2\Omega} \sum_{\mathbf{k},\eta} \left[ u^{\uparrow}_{\mathbf{k}-,\eta} \left(v^{\downarrow}_{\mathbf{k}-,\eta}\right)^* + u^{\downarrow}_{\mathbf{k}+,\eta} \left(v^{\uparrow}_{\mathbf{k}+,\eta}\right)^* \right], \tag{11}$$

where $\Omega$ denotes the system volume (here: area). As the quasiparticle excitation energies and associated spinor amplitudes are themselves functions of $\Delta$, Eq. (11) constitutes a self-consistency condition [35, 38]. However, the expression (11) is formally divergent and needs to be regularized using the relation [48]

$$\frac{1}{g} = -\frac{1}{\Omega} \sum_{\mathbf{k}} \frac{1}{2\epsilon_{\mathbf{k}} + E_{\mathrm{b}}}, \tag{12}$$

where $E_b > 0$ is the absolute value of the binding energy of the two-particle bound state in 2D [52–54] in the absence of spin-orbit coupling, i.e., for $\lambda = 0$. (Modifications of the two-body bound state in a quasi-2D Fermi gas due to spin-orbit coupling are discussed in Refs. [50, 55, 56], but these are not relevant for the pairing-gap regularisation procedure [57].) The binding energy is related to the 2D scattering length via the expression $E_b = 4e^{-2\gamma}\hbar^2/(ma_{2D}^2)$, where $\gamma = 0.577\ldots$ is the Euler constant[4]. Recent experimental realizations of low-temperature 2D Fermi gases [61–63] have been able to access a wide parameter range $-7 \lesssim \ln(k_F a_{2D}) \lesssim 4$. Combination of Eqs. (11) and (12) yields the practically relevant $s$-wave pair-potential self-consistency condition

$$0 = \frac{1}{\Omega}\sum_{\mathbf{k}}\left\{\frac{1}{\Delta}\sum_{\eta}\left[u_{\mathbf{k}-,\eta}^{\uparrow}\left(v_{\mathbf{k}-,\eta}^{\downarrow}\right)^{*} + u_{\mathbf{k}+,\eta}^{\downarrow}\left(v_{\mathbf{k}+,\eta}^{\uparrow}\right)^{*}\right] - \frac{2}{2\epsilon_{\mathbf{k}} + E_b}\right\}. \tag{13}$$

The densities $n_\sigma$ of spin-$\sigma$ particles are also implicit functions of system parameters via the expressions

$$n_\sigma = \frac{1}{\Omega}\sum_{\mathbf{k}} n_{\mathbf{k}\sigma}, \tag{14a}$$

$$n_{\mathbf{k}\sigma} = \frac{1}{2}\sum_{\eta}\left(\left|u_{\mathbf{k}-,\eta}^{\sigma}\right|^2 + \left|v_{\mathbf{k}+,\eta}^{\sigma}\right|^2\right). \tag{14b}$$

For a uniform Fermi gas with fixed total particle number density $n \equiv \sum_\sigma n_\sigma$, we thus have a second self-consistency condition given by

$$1 = \frac{1}{\Omega}\sum_{\mathbf{k}}\frac{1}{n}\sum_{\sigma} n_{\mathbf{k}\sigma}. \tag{15}$$

Explicit analytical expressions for the momentum-space density distributions $n_{\mathbf{k}\sigma}$ defined in (14b) and the quantity

$$\Upsilon_{\mathbf{k}} = \frac{1}{\Delta}\sum_{\eta}\left[u_{\mathbf{k}-,\eta}^{\uparrow}\left(v_{\mathbf{k}-,\eta}^{\downarrow}\right)^{*} + u_{\mathbf{k}+,\eta}^{\downarrow}\left(v_{\mathbf{k}+,\eta}^{\uparrow}\right)^{*}\right] \tag{16}$$

entering the r.h.s. of Eq. (13) have been derived within the exact $4 \times 4$ BdG theory [27, 33]. We now discuss in some detail the corresponding results provided by the approximate $2 \times 2$-projected approach developed here.

## 3.1 Momentum-space density distributions and chemical potential

Using results for the spinor amplitudes given in Eq. (8b), we obtain the momentum-space distribution $n_{\mathbf{k}\sigma}$ of the spin-$\sigma$ particle density as a sum of contributions from the two projected $2 \times 2$ sectors, $n_{\mathbf{k}\sigma} = n_{\mathbf{k}\sigma}^{\sigma} + n_{\mathbf{k}\sigma}^{\bar{\sigma}}$, where

$$n_{\mathbf{k}\sigma}^{\sigma} = \frac{N_{\mathbf{k}+}^{\sigma} + N_{\mathbf{k}-}^{\sigma}}{4}\left(1 - \frac{\xi_{\mathbf{k}\sigma}}{E_{\mathbf{k}+}^{\sigma}}\right), \tag{17a}$$

$$n_{\mathbf{k}\sigma}^{\bar{\sigma}} = \frac{N_{\mathbf{k}+}^{\bar{\sigma}} + N_{\mathbf{k}-}^{\bar{\sigma}}}{4}\frac{|\Delta|^2 + |\lambda_{\mathbf{k}}|^2 + (|\Delta|^2 - |\lambda_{\mathbf{k}}|^2)(\xi_{\mathbf{k}\bar{\sigma}}/E_{\mathbf{k}+}^{\bar{\sigma}})\,_{(+)}^{-}\,2|\lambda_{\mathbf{k}}|^2|\Delta|^2/(h_{\mathbf{k}}E_{\mathbf{k}+}^{\bar{\sigma}})}{\left(E_{\mathbf{k}+}^{\bar{\sigma}} + \epsilon_{\mathbf{k}\sigma} - \mu\right)^2}, \tag{17b}$$

---

[4]Generally, $E_b \sim \hbar^2/(ma_{2D}^2)$ for shallow dimers, but values given in the literature for the prefactor on the r.h.s. of that relation vary. This is due to different conventions being used when defining the two-dimensional scattering length $a_{2D}$ [58–60]. Our choice follows related previous work [33, 34].

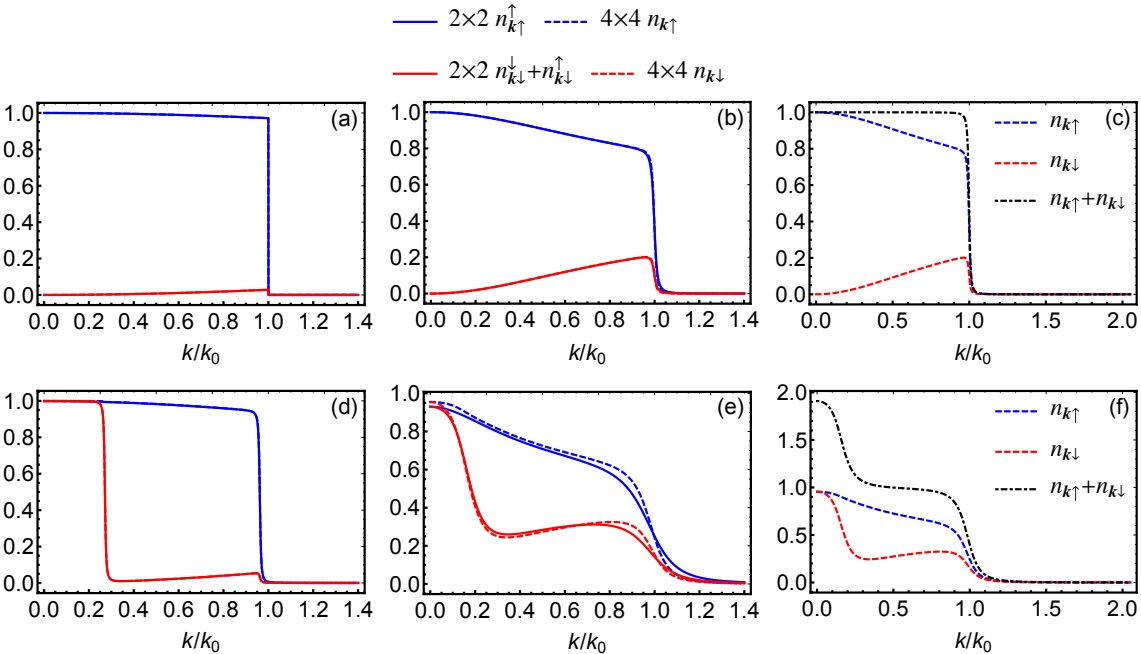

Figure 3: Spin-$\sigma$ particle-density distributions $n_{\mathbf{k}\sigma}$ in topological superfluids [panels (a-c)] and non-topological superfluids [panels (d-f)]. Curves labeled $2 \times 2$ ($4 \times 4$) are obtained within our $2 \times 2$ projected theory, omitting the contribution $n_{\mathbf{k}\uparrow}^{\downarrow}$ to $n_{\mathbf{k}\uparrow}$ (the exact $4 \times 4$ approach). The energy and momentum scales $E_0$ and $k_0$ are related via $E_0 \equiv \hbar^2 k_0^2/(2m)$. Panels (a) and (d) show situations with excellent agreement between the two approaches. Values for relevant parameters are $2m\lambda/\hbar^2 k_0 = 0.21$ [in both panels], $h/E_0 = 0.60$ [in (a)] 0.40 [in (d)], $\mu/E_0 = 0.36$ [in (a)] 0.48 [in (d)], and $|\Delta|/E_0 = 6.3 \times 10^{-5}$ [in (a)] 0.024 [in (d)], which correspond to self-consistent results obtained for $E_{\mathrm{b}}/E_{\mathrm{F}} = 0.50$ when setting $k_0 = \sqrt{2}\,k_{\mathrm{F}}$. Deviations between $2 \times 2$ and $4 \times 4$ results occur for larger spin-orbit-coupling strength and are more pronounced in the non-topological regime, as illustrated in panels (b) and (e). Here $2m\lambda/(\hbar^2 k_0) = 0.71$ [in both (b) and (e)], $h/E_0 = 0.50$ [in (b)] 0.20 [in (e)], $\mu/E_0 = 0.13$ [in (b)] 0.24 [in (e)], and $|\Delta|/E_0 = 0.019$ [in (b)] 0.11 [in (e)]. With $k_0 = \sqrt{2}\,k_{\mathrm{F}}$, these are the parameters obtained self-consistently for $E_{\mathrm{b}}/E_{\mathrm{F}} = 0.050$. Panel (c) [(f)] again displays the exactly calculated density distributions $n_{\mathbf{k}\sigma}$ from panel (b) [(e)], with their sum also shown as the dot-dashed curve.

and the upper (lower) sign of the last term in the numerator of Eq. (17b) applies to $\sigma = \uparrow$ ($\downarrow$). Interestingly, $n_{\mathbf{k}\uparrow}^{\downarrow}$ turns out to be negligible except for an unphysical divergence at the point where the expression in the denominator of Eq. (17b) for $\sigma = \uparrow$ vanishes. This artefact of our approximations is remedied by neglecting $n_{\mathbf{k}\uparrow}^{\downarrow}$ (i.e., setting $n_{\mathbf{k}\uparrow} \equiv n_{\mathbf{k}\uparrow}^{\uparrow}$ within the $2 \times 2$-projected theory) from now on. In contrast, $n_{\mathbf{k}\downarrow}^{\uparrow}$ is well-behaved [as the denominator of Eq. (17b) for $\sigma = \downarrow$ is always finite] and contributes importantly to $n_{\mathbf{k}\downarrow}$. Figure 3 shows a comparison between the density distributions $n_{\mathbf{k}\sigma}$ thus obtained within the $2 \times 2$-projected theory with those calculated within the exact $4 \times 4$ formalism[5]. There is excellent agreement for the spin-resolved density distributions from both approaches as long as spin-orbit coupling is not too strong. For fixed spin-orbit-coupling strength, deviations are greater for smaller values of the Zeeman splitting $h$, i.e., these tend to be more pronounced in the non-topological regime.

Interestingly, the concept of separate spin-$\downarrow$ and spin-$\uparrow$ Fermi spheres with radius $k_\downarrow$ and

---

[5]Curves corresponding to exact results obtained from $4 \times 4$ theory in Fig. 3 can be also usefully compared with the momentum-space density distributions of a 3D Fermi gas with 2D Rashba spin-orbit coupling calculated for fixed $k_z = 0$. Pertinent results are shown, e.g., as insets of Figs. 3(c) and 3(d) in Ref. [25].

$k_\uparrow$, respectively, turns out to be useful even at significant levels of spin-orbit coupling, since the exact density distributions satisfy very accurately the approximate relation

$$n_{\mathbf{k}\uparrow} + n_{\mathbf{k}\downarrow} \approx \Theta(k_\uparrow - k) + \Theta(k_\downarrow - k), \tag{18}$$

as is illustrated in Figs. 3(c) and 3(f). Here $\Theta(\cdot)$ denotes the Heaviside step function, $k_\uparrow > k_\downarrow$ generically, and $k_\downarrow \equiv 0$ in the topological regime. Motivated by observing the apparent broad validity of relation (18), we insert it into the number-density equation (15) and straightforwardly derive the result

$$2k_F^2 = k_\uparrow^2 + k_\downarrow^2 \tag{19}$$

as an equivalent self-consistency condition. Furthermore, Eq. (2) together with the fact that $E_{\mathbf{k}+,<} \approx 0$ for $|\mathbf{k}| = k_\uparrow$ (generally valid to leading order in small $|\Delta|$) implies

$$\mu \approx \frac{\hbar^2 k_\uparrow^2}{2m} - \sqrt{h^2 + \lambda^2 k_\uparrow^2}. \tag{20a}$$

For the case where $k_\downarrow \neq 0$, we can extrapolate Eq. (2) to the point $E_{\mathbf{k}+,>} \approx 0$ when $|\mathbf{k}| = k_\downarrow$ and find

$$\mu \approx \frac{\hbar^2 k_\downarrow^2}{2m} + \sqrt{h^2 + \lambda^2 k_\downarrow^2} \qquad (k_\downarrow > 0). \tag{20b}$$

In the topological regime (realized for $h > h_c$, where $h_c$ denotes the value for the Zeeman energy at the transition), $k_\downarrow = 0$ so that Eq. (19) implies $k_\uparrow = \sqrt{2}\,k_F$ and (20a) yields the approximate relation

$$\frac{\mu}{E_F} \approx 2 - \sqrt{\left(\frac{h}{E_F}\right)^2 + 2\left(\frac{2m\lambda}{\hbar^2 k_F}\right)^2} \qquad (h > h_c) \tag{21}$$

between chemical potential and particle density. In the non-topological regime (realized for $h < h_c$), $k_\downarrow \neq 0$ and we need to simultaneously solve Eqs. (20a), (20b) and (19). Adding (20b) to (20a), using (19), and expanding to first sub-leading order in large $h$, we find

$$\frac{\mu}{E_F} \approx 1 - \frac{m\lambda^2}{2\hbar^2 h} \frac{k_\uparrow^2 - k_\downarrow^2}{k_F^2} \qquad (h < h_c). \tag{22a}$$

Furthermore, subtracting (20b) from (20a) and expanding again to first sub-leading order in large $h$ yields

$$\frac{k_\uparrow^2 - k_\downarrow^2}{k_F^2} \approx \frac{2h}{E_F} + \frac{2m\lambda^2}{\hbar^2 h} \qquad (h < h_c), \tag{22b}$$

where we have again also used Eq. (19). Combining the results from Eqs. (22a) and (22b), we find the relation

$$\frac{\mu}{E_F} \approx 1 - \frac{1}{2}\left(\frac{2m\lambda}{\hbar^2 k_F}\right)^2 \qquad (h < h_c) \tag{23}$$

between chemical potential and particle number that is valid in the non-topological regime, to leading order in large $h$.

Figure 4 shows a detailed comparison between the self-consistent chemical potential obtained from the exact $4 \times 4$ approach, from the effective $2 \times 2$-projected theory developed here, and the approximate analytical expressions (21) and (23). The situation depicted in panel (a) is the same as in Fig. 1(b) of Ref. [33]. Capitalizing on the weak $|\Delta|$ dependence, we used fixed values for the $s$-wave gap in our calculation, corresponding to the self-consistent results at $h = h_c$ for $E_b/E_F = 0.0462$. The curve for the $2 \times 2$-projected theory is seen to agree very well

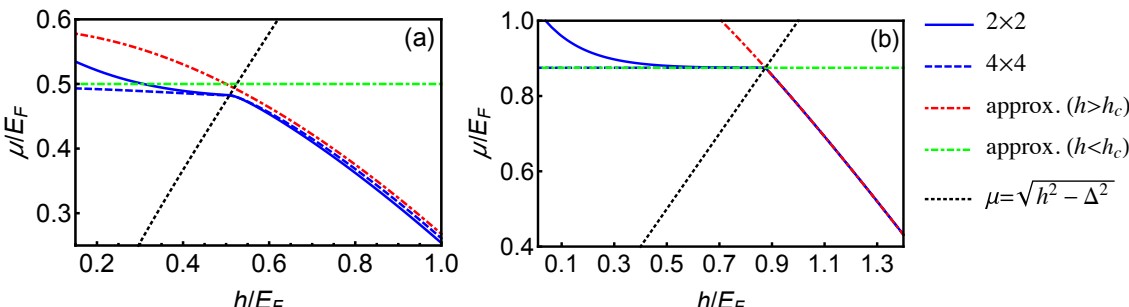

Figure 4: Variation of the chemical potential $\mu$ with Zeeman energy $h$ for fixed total density $n = mE_{\rm F}/(\pi\hbar^2)$. Curves labeled $2\times 2$ ($4\times 4$) are obtained using our $2\times 2$-projected theory, omitting the contribution $n_{\mathbf{k}\uparrow}^{\downarrow}$ to $n_{\mathbf{k}\uparrow}$ (the exact $4\times 4$ approach). Panel (a) [(b)] shows results calculated for $2m\lambda/(\hbar^2 k_{\rm F}) = 1.00$ [0.500]. For convenience, we fixed $|\Delta|/E_{\rm F} = 0.159$ [$3.48\times 10^{-4}$] in the calculation, which is the self-consistent value at the critical Zeeman energy $h_{\rm c}/E_{\rm F} = 0.507$ [0.875] for $E_{\rm b}/E_{\rm F} = 0.0462$, i.e., $\ln(k_{\rm F}a_{\rm 2D}) = 2.00$. The approximate analytical formulae from Eqs. (21) and (23) are plotted as the dot-dashed curves, and the dotted curve indicates the condition for the transition between non-topological and topological superfluid phases.

with the exact result for large-enough $h/E_{\rm F}$, which includes not only the topological regime but also part of the non-topological regime near the transition. Deviations between the $2\times 2$-projected theory and the exact $4\times 4$ results become significant in the limit of small $h$, where the Feshbach-projection approach is indeed expected to fail. As illustrated by the situation shown in panel (b), the agreement between the effective-$2\times 2$ and exact-$4\times 4$ results becomes excellent for smaller values of $\lambda$, reaching also deeper into the non-topological regime. The observation that the exact $4\times 4$-theory results for $\mu/E_{\rm F}$ and the approximate analytical expressions given in Eqs. (21) and (23) are practically indistinguishable at small-enough magnitude of spin-orbit coupling suggests the possibility to utilize these analytical formulae, for better efficiency and greater insight, as input into the self-consistent calculation of the $s$-wave pair potential. The analytical expression (23) could also be useful to more accurately represent the chemical potential in the low-$h$ limit where the $2\times 2$-projected results deviate significantly from those obtained from the exact $4\times 4$ approach.

## 3.2 Self-consistency of $s$-wave pair potential: Spin-$\uparrow$-projected theory

The approximate description based on the $2\times 2$-subspace projections gave Eq. (8b) for the Nambu-spinor amplitudes. Inserting these expressions into (16) yields $\Upsilon_{\mathbf{k}} = \Upsilon_{\mathbf{k}}^{\uparrow} + \Upsilon_{\mathbf{k}}^{\downarrow}$, with

$$\Upsilon_{\mathbf{k}}^{\uparrow(\downarrow)} = \frac{N_{\mathbf{k}+}^{\uparrow(\downarrow)} \overset{+}{(-)} N_{\mathbf{k}-}^{\uparrow(\downarrow)}}{2E_{\mathbf{k}+}^{\uparrow(\downarrow)}} \frac{\epsilon_{\mathbf{k}\uparrow(\downarrow)} - \mu \overset{-}{(+)} E_{\mathbf{k}+}^{\uparrow(\downarrow)} \overset{+}{(-)} (|\Delta|^2 + |\lambda_{\mathbf{k}}|^2)/(2h_{\mathbf{k}})}{\epsilon_{\mathbf{k}\downarrow(\uparrow)} - \mu \overset{-}{(+)} E_{\mathbf{k}+}^{\uparrow(\downarrow)}}. \tag{24}$$

Using our approximation (10) for the normalization factors and, for consistency, replacing $\epsilon_{\mathbf{k}\downarrow} - \mu - E_{\mathbf{k}+}^{\uparrow} \approx 2h_{\mathbf{k}}$ in the denominator of $\Upsilon_{\mathbf{k}}^{\uparrow}$, we obtain

$$\Upsilon_{\mathbf{k}}^{\uparrow} \approx \frac{1}{E_{\mathbf{k}+}^{\uparrow}} \frac{2h_{\mathbf{k}}\left(\epsilon_{\mathbf{k}\uparrow} - \mu - E_{\mathbf{k}+}^{\uparrow}\right) + |\Delta|^2 + |\lambda_{\mathbf{k}}|^2}{4h_{\mathbf{k}}^2 + |\Delta|^2 + |\lambda_{\mathbf{k}}|^2}, \tag{25a}$$

$$\Upsilon_{\mathbf{k}}^{\downarrow} \approx 0. \tag{25b}$$

Hence, we find that the self-consistency condition for the $s$-wave pair potential can be formulated entirely in terms of quantities relating to the projected spin-$\uparrow$ degrees of freedom.

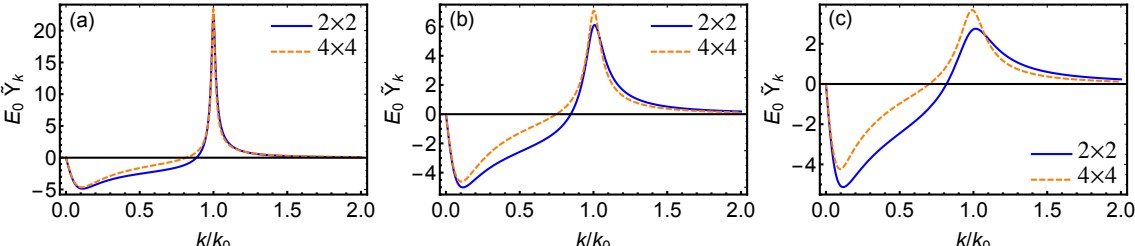

Figure 5: Line shape of the summand $\tilde{\Upsilon}_{\mathbf{k}} = \Upsilon_{\mathbf{k}} - 2/(2\epsilon_{\mathbf{k}} + E_{\mathrm{b}})$ in the self-consistency condition (13) for the $s$-wave pair potential. Curves labeled $2 \times 2$ ($4 \times 4$) are calculated from the $2 \times 2$-projected theory using $\Upsilon_{\mathbf{k}} \equiv \Upsilon_{\mathbf{k}}^{\uparrow}$ with (25a) (from the exact $4 \times 4$-theory expression for $\tilde{\Upsilon}_{\mathbf{k}}$). The parameters $E_{\mathrm{b}}/E_0 = 0.023$ and $2m\lambda/(\hbar^2 k_0) = 0.71$ are fixed in all panels, whereas $h/E_0 = 0.50$ [0.30, 0.20], $\mu/E_0 = 0.13$ [0.23, 0.24], and $|\Delta|/E_0 = 0.017$ [0.060, 0.11] for panel (a) [(b), (c)]. With $k_0 = \sqrt{2}\,k_{\mathrm{F}}$, these values coincide with those used for/obtained by numerical calculations whose results are shown in Fig. 1 of Ref. [33]. The energy and momentum scales $E_0$ and $k_0$ are related via $E_0 \equiv \hbar^2 k_0^2/(2m)$.

Figure 5 shows the $k$ dependence of the quantity $\tilde{\Upsilon}_{\mathbf{k}} = \Upsilon_{\mathbf{k}} - 2/(2\epsilon_{\mathbf{k}} + E_{\mathrm{b}})$ that is the summand in the self-consistency equation (13) for the $s$-wave pair potential. Parameters are chosen to coincide with those from a recent numerical study [33]. The system is deep in the topological-superfluid regime for panel (a), still topological but close to the transition in panel (b), and a non-topological superfluid close to the transition in panel (c). Perhaps not surprisingly, the agreement between the projected $2 \times 2$ theory and the exact $4 \times 4$ formalism is best deep in the topological-superfluid regime, as the fidelity of the projected theory should improve for increasing $h$. Generally, the small-$k$ and large-$k$ behaviors of $\tilde{\Upsilon}_{\mathbf{k}}$ are captured almost perfectly within the projected $2 \times 2$ theory, with deviations at smaller $h$ occurring chiefly at intermediate values of $k$. However, as the self-consistency condition (13) involves a sum over all $k$, the overall effect of such deviations cannot be easily ascertained without explicitly finding the self-consistent $s$-wave pair potentials within both the $4 \times 4$ and $2 \times 2$ approaches. Such a detailed comparison is one of the foci of the next Section.

# 4 Superfluidity with uniform *s*-wave pair potential: Effective two-band description *versus* exact four-band theory

The complete description of superfluidity for a uniform system in the experimentally relevant situation with fixed total particle density requires the simultaneous solution of the self-consistency conditions (13) and (15). This is generally achieved by standard iterative procedures that are based on a fully explicit knowledge of the exact four-band spectrum $E_{\mathbf{k}\alpha,\eta}$ and the associated eigenspinors. Although such a procedure has the advantage of yielding exact results, its complicated formal structure obscures possibilities for gaining a deeper intuitive understanding of relevant physical effects. In contrast, the formalism developed in Sec. 3 offers the attractive alternative to be able to describe the system entirely in terms of a conceptually simpler theory based on the spin-↑-projected (two-band) spectrum. We now investigate in greater detail the physical picture provided by the effective two-band approach, where the self-consistency condition (13) is solved using $\Upsilon_{\mathbf{k}} \equiv \Upsilon_{\mathbf{k}}^{\uparrow}$ with (25a) and approximating the chemical potential by Eqs. (21) and (23) as appropriate.

## 4.1 Boundary between topological and non-topological phases

We start by considering relevant thermodynamic quantities at the transition between the non-topological and topological superfluid regimes. This transition occurs at the value $h \equiv h_c$ of the Zeeman energy that satisfies the condition

$$h_c = \sqrt{\mu^2 + |\Delta|^2}. \tag{26}$$

For a given system with fixed uniform total particle density $n$ and $s$-wave interaction strength measured in terms of the two-body bound-state energy $E_b$, both $\mu$ and $|\Delta|$ are implicit functions of $h$ and $n$ via the self-consistency conditions and, thus, their values $\mu_c \equiv \mu(h_c)$ and $\Delta_c \equiv |\Delta(h_c)|$ are also fixed. In Table 1, we summarize these critical values obtained using the exact $4 \times 4$ theory and compare with those calculated within the effective $2 \times 2$ approach using two different methods. To obtain $\mu_c^{2\times2}$ and $\Delta_c^{2\times2}$, we simultaneously solve the self-consistency conditions for the number density and pair potential, Eqs. (15) and (13), assuming also $n_{\mathbf{k}\uparrow} \equiv n_{\mathbf{k}\uparrow}^{\uparrow}$, $n_{\mathbf{k}\downarrow} \equiv n_{\mathbf{k}\downarrow}^{\downarrow} + n_{\mathbf{k}\downarrow}^{\uparrow}$, and $\Upsilon_{\mathbf{k}} \equiv \Upsilon_{\mathbf{k}}^{\uparrow}$ with relevant expressions given in Eqs. (17a), (17b), and (25a). In contrast, $\tilde{\Delta}_c^{2\times2}$ is the result of a simpler routine where only the pair-potential self-consistency condition (13) is solved, setting $\Upsilon_{\mathbf{k}} \equiv \Upsilon_{\mathbf{k}}^{\uparrow}$ with Eq. (25a) and approximating $\mu_c/E_F$ by Eq. (23).

Inspection of Table 1 shows that the values obtained for $\Delta_c^{2\times2}$ and $\tilde{\Delta}_c^{2\times2}$ are generally very close, even in the regime where the approximation (23) for $\mu_c/E_F$ is not accurate. [For easy reference, values for $\mu_c/E_F$ and $\mu_c^{2\times2}/E_F$ that agree to within 5% with the analytical approximation Eq. (23) are indicated in green.] Thus, at least to determine critical values within the effective $2 \times 2$-projected theory, using the simpler routine yielding $\tilde{\Delta}_c^{2\times2}$ is a viable approach. Interestingly, the agreement between values for $\Delta_c$ and $\tilde{\Delta}_c^{2\times2}$ turns out to be generally better for larger $\lambda$. (Values for $\tilde{\Delta}_c^{2\times2}$ indicated in magenta are close to within 25% to the exact $4 \times 4$ results.) More specifically, even though the assumption (23) made for $\mu$ when determining $\tilde{\Delta}_c$ is more broadly valid across the range of accessible $E_b$ at smaller $\lambda$, the projected $2 \times 2$-theory's self-consistency equations appear to fail for small $|\Delta|$. As a rule of thumb, the condition $2m\lambda/(\hbar^2 k_F) \gtrsim 1$ is needed for $2 \times 2$-theory results to be in reasonable agreement with the exact $4 \times 4$ values $\Delta_c$. Surprisingly, at larger $\lambda$, the rather good agreement between $\Delta_c$ and $\tilde{\Delta}_c^{2\times2}$ extends even to situations where $\mu_c$ differs significantly from the approximation (23).

In the regime of small $|\Delta|$, for which Fig. 4(b) is an illustration, the approximations Eqs. (21) and (23) are accurate over the entire range of Zeeman energies $h$, including the critical value $h_c$ where both expression yield coinciding values. Thus, from the condition that the right-hand sides of (21) and (23) are equal, we can obtain an approximate expression for the phase boundary in $h$–$\lambda$ space,

$$\frac{h_c}{E_F} \approx \left| 1 - \frac{1}{2} \left( \frac{2m\lambda}{\hbar^2 k_F} \right)^2 \right| \qquad (|\Delta| \ll \mu). \tag{27}$$

The result (27) is consistent with the expectation that $h_c \approx |\mu_c|$ for $|\Delta| \ll \mu$, which follows straightforwardly from (26), in conjunction with the validity of the approximation (23). Figure 6 shows the phase boundary calculated within the projected $2 \times 2$ theory by solving the self-consistency condition (13) by setting $\Upsilon_{\mathbf{k}} \equiv \Upsilon_{\mathbf{k}}^{\uparrow}$ with (25a) and approximating $\mu/E_F$ by (23) while also enforcing the relation (26). For comparison, the approximation (27) and exact results obtained from the $4 \times 4$ formalism are also included in these plots. [The phase boundary found within the $2 \times 2$-projected theory by simultaneously self-consistent determination of $\Delta_c^{2\times2}$ and $\mu_c^{2\times2}$ differs only imperceptibly from the more easily obtained $2 \times 2$-theory curve shown in Fig. 6 where $\mu_c/E_F$ is approximated by (23) in the calculation of the critical

Table 1: Chemical potential $\mu(h_c) \equiv \mu_c$ and $s$-wave gap $|\Delta(h_c)| \equiv \Delta_c$ at the critical Zeeman energy $h_c$ where the transition between topological and non-topological regimes occurs, calculated exactly within $4 \times 4$ theory and compared with results from the effective $2 \times 2$ approach ($\mu_c^{2\times2}$, $\Delta_c^{2\times2}$). The value $\tilde{\Delta}_c^{2\times2}$ is the critical gap obtained from $2 \times 2$ theory when $\mu_c/E_F$ is approximated by Eq. (23). Values for $\mu_c$ that agree with Eq. (23) to within 5% are shown in green. Results for $\tilde{\Delta}_c^{2\times2}$ given in magenta agree with $\Delta_c$ to within 25%.

| $2m\lambda/(\hbar^2 k_F)$ | $\ln(k_F a_{2D})$ | 0.500 | 1.00 | 1.50 | 2.00 | 2.50 | 3.00 |
|---|---|---|---|---|---|---|---|
| | $E_b/E_F$ | 0.928 | 0.341 | 0.126 | 0.0462 | 0.0170 | 0.00625 |
| 1.50 | $h_c/E_F$ | 1.32 | 0.802 | 0.524 | 0.357 | 0.253 | 0.189 |
| | $\mu_c/E_F$ | −0.660 | −0.323 | −0.188 | −0.137 | −0.120 | −0.117 |
| | $\Delta_c/E_F$ | 1.14 | 0.735 | 0.490 | 0.330 | 0.222 | 0.148 |
| | $\mu_c^{2\times2}/E_F$ | −0.348 | −0.207 | −0.132 | −0.0943 | −0.0782 | −0.0743 |
| | $\Delta_c^{2\times2}/E_F$ | 0.775 | 0.582 | 0.444 | 0.344 | 0.269 | 0.213 |
| | $\tilde{\Delta}_c^{2\times2}/E_F$ | 0.792 | 0.586 | 0.445 | 0.343 | 0.269 | 0.212 |
| 1.25 | $h_c/E_F$ | 1.15 | 0.689 | 0.457 | 0.331 | 0.266 | 0.236 |
| | $\mu_c/E_F$ | −0.291 | 0.0234 | 0.144 | 0.190 | 0.208 | 0.214 |
| | $\Delta_c/E_F$ | 1.11 | 0.689 | 0.434 | 0.271 | 0.166 | 0.0982 |
| | $\mu_c^{2\times2}/E_F$ | 0.0940 | 0.173 | 0.207 | 0.221 | 0.225 | 0.225 |
| | $\Delta_c^{2\times2}/E_F$ | 0.609 | 0.433 | 0.313 | 0.227 | 0.166 | 0.122 |
| | $\tilde{\Delta}_c^{2\times2}/E_F$ | 0.590 | 0.425 | 0.310 | 0.228 | 0.168 | 0.124 |
| 1.00 | $h_c/E_F$ | 1.07 | 0.685 | 0.546 | 0.507 | 0.501 | 0.500 |
| | $\mu_c/E_F$ | 0.0291 | 0.330 | 0.442 | 0.482 | 0.495 | 0.499 |
| | $\Delta_c/E_F$ | 1.07 | 0.600 | 0.320 | 0.159 | 0.0758 | 0.0358 |
| | $\mu_c^{2\times2}/E_F$ | 0.439 | 0.472 | 0.487 | 0.494 | 0.498 | 0.499 |
| | $\Delta_c^{2\times2}/E_F$ | 0.352 | 0.215 | 0.131 | 0.0794 | 0.0481 | 0.0291 |
| | $\tilde{\Delta}_c^{2\times2}/E_F$ | 0.329 | 0.206 | 0.128 | 0.0782 | 0.0477 | 0.0290 |
| 0.75 | $h_c/E_F$ | 1.04 | 0.747 | 0.718 | 0.718 | 0.719 | 0.719 |
| | $\mu_c/E_F$ | 0.317 | 0.630 | 0.707 | 0.717 | 0.719 | 0.719 |
| | $\Delta_c/E_F$ | 0.990 | 0.401 | 0.127 | 0.0401 | 0.0128 | 0.00409 |
| | $\mu_c^{2\times2}/E_F$ | 0.713 | 0.718 | 0.718 | 0.719 | 0.719 | 0.719 |
| | $\Delta_c^{2\times2}/E_F$ | 0.0702 | 0.0287 | 0.0118 | 0.00484 | 0.00199 | 0.000818 |
| | $\tilde{\Delta}_c^{2\times2}/E_F$ | 0.0687 | 0.0285 | 0.0118 | 0.00484 | 0.00199 | 0.000818 |

Zeeman energy.] We restrict ourselves to showing the phase boundary only for intermediate values of $2m\lambda/(\hbar^2 k_F)$ where superfluidity is not expected to be destabilized by phase separation [27, 32].

Interestingly, the approximated $2 \times 2$ approach turns out to predict the phase boundary between topological and non-topological phases correctly over a broader range of spin-orbit-coupling strengths than naïvely expected when considering the deviations between $\Delta_c$ and $\tilde{\Delta}_c^{2\times2}$ given in Table 1. This is the result of $h_c$ being generally dominated either by the value of $\mu_c$ or that of $\Delta_c$. Although the $2 \times 2$-projected theory significantly underestimates $\Delta_c$ for small $\lambda$, $h_c$ is dominated by the chemical potential in this parameter range, in which the expression (23) for $\mu_c$ is highly accurate. On the other hand, for large-enough $\lambda$ when $\Delta_c$ starts to become more important than $\mu_c$ for determining $h_c$, the $2 \times 2$ approach yields quite accurate values for $\Delta$. As a result, the $h_c(\lambda)$ dependence obtained within the projected $2 \times 2$ theory faithfully reproduces known qualitative features such as the minimum at $2m\lambda/(\hbar^2 k_F) \gtrsim 1$ [32].

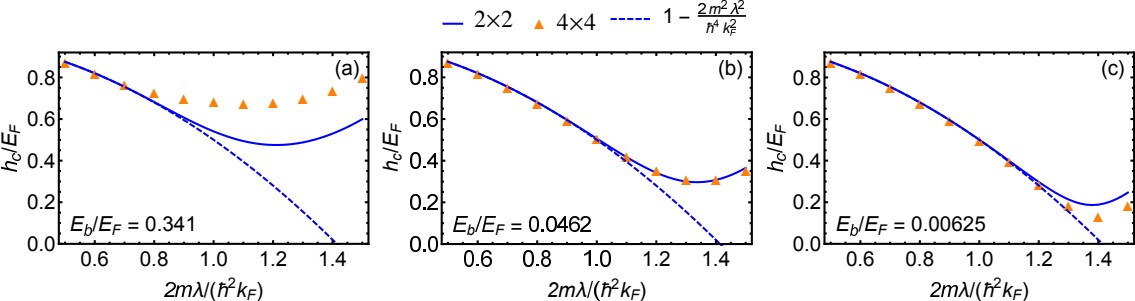

Figure 6: Phase boundary between topological and non-topological superfluid states that occur for $h > h_\mathrm{c}$ and $h < h_\mathrm{c}$, respectively. Results labeled $2 \times 2$ ($4 \times 4$) were calculated by finding the Zeeman energy satisfying (26) from solution of the self-consistency equation (13) for the $s$-wave pair potential using $\Upsilon_\mathbf{k} \equiv \Upsilon_\mathbf{k}^\uparrow$ with (25a) and approximating the chemical potential $\mu$ by (23) (by simultaneous solution of the exact $4 \times 4$-theory self-consistency equations for $\Delta$ and $\mu$), using the indicated values for $E_\mathrm{b}/E_\mathrm{F}$. The latter correspond to $\ln(k_\mathrm{F} a_\mathrm{2D}) = 1.00, 2.00, 3.00$, respectively (cf. Table 1). Dashed curves show the approximate expression (27) that is expected to be valid in the regime where $|\Delta| \ll \mu$.

## 4.2 Parametric dependences of the self-consistent $s$-wave pair potential

The magnitude $|\Delta|$ of the $s$-wave pair potential depends intricately on the tunable system parameters $n, h, \lambda,$ and $E_\mathrm{b}$ through the self-consistency conditions (13) and (15). As it turns out, the dependence on the particle density $n$ is most conveniently absorbed by using the Fermi wave vector $k_\mathrm{F}$ and Fermi energy $E_\mathrm{F}$ as units for all other quantities to be measured in. Figure 7 illustrates the $\lambda$ and $h$ dependence of $|\Delta|$ and provides a comparison between results obtained within the approximate $2 \times 2$ approach and the exact $4 \times 4$ theory. [Both the simplified $2 \times 2$-theory self-consistency routine where $\mu_\mathrm{c}/E_\mathrm{F}$ is approximated by (23) and the simultaneously self-consistent determination of $\mu$ and $|\Delta|$ within the $2 \times 2$-projected approach yield practically indistinguishable results for the parameters chosen in the Figure.] We show numbers pertaining to fixed $E_\mathrm{b}/E_\mathrm{F} = 0.0462$, corresponding to $\ln(k_\mathrm{F} a_\mathrm{2D}) = 2$, to enable direct comparison also with previous works [33, 34] that give numerical results for $|\Delta|$ vs. $h$ [6].

From the derivation of the main decoupling approximation (5) of the projected approach, we may expect good agreement between the $2 \times 2$ and $4 \times 4$ results when $|\Delta| \ll h$, which is generally supported by the results reported in Fig. 7. It should be noted, though, that obtaining a small $|\Delta|$ from the self-consistent $2 \times 2$ theory is not sufficient to guarantee this situation, as can be seen from Fig. 7(a), where $|\Delta|/h > 1$ according to the $4 \times 4$ equations but the accidental compensation of positive and negative parts of the summand (as shown in Fig. 5) results in small values for $|\Delta|$ within the approximate $2 \times 2$ theory. In such situations, the projected $2 \times 2$ approach typically tends to underestimate the value of the self-consistent $|\Delta|$.

Figure 7(a) [7(b)] shows the $\lambda$ dependence of $|\Delta|$ for a situation where the system is in the non-topological [topological] superfluid phase. The same qualitative behavior of $|\Delta|$ increasing for increased $\lambda$ is exhibited in both panels (a) and (b), for both the $2 \times 2$ and $4 \times 4$ data points. However, the much weaker $|\Delta|$-vs.-$\lambda$ dependence in the non-topological phase is not reproduced correctly by the approximate $2 \times 2$ formalism, whereas there is quite good agreement with the exact results in the topological phase. This is expected, as the projected

---

[6]The $|\Delta|$-vs.-$\lambda$ dependence for a 2D Fermi superfluid was explored before in Ref. [32], albeit for a situation where the spin polarization $(n_\uparrow - n_\downarrow)/n$ was held fixed instead of the Zeeman energy $h$. However, as can be seen from Fig. 4 in that work, $h$ turns out to be effectively constant in the range $2m\lambda/(\hbar^2 k_\mathrm{F}) \gtrsim 0.5$ relevant for our present study. Thus we can safely compare our results for the $|\Delta|$-vs.-$\lambda$ dependence, at the very least its qualitative behavior, with that presented in Ref. [32].

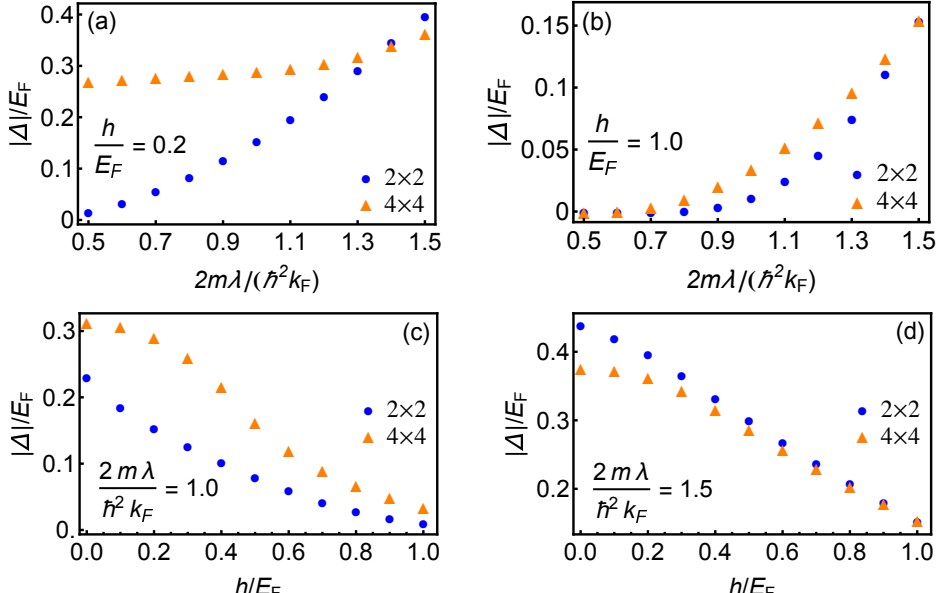

Figure 7: Magnitude $|\Delta|$ of the $s$-wave pair potential obtained self-consistently as a function of spin-orbit coupling strength $\lambda$ and Zeeman energy $h$. Data points labeled $2 \times 2$ ($4 \times 4$) were calculated by solving the self-consistency equation (13) for the $s$-wave pair potential using $\Upsilon_{\mathbf{k}} \equiv \Upsilon_{\mathbf{k}}^{\uparrow}$ with (25a) and approximating the chemical potential $\mu$ by Eqs. (21) and (23) as appropriate (by simultaneous solution of the exact $4 \times 4$-theory self-consistency equations for $\Delta$ and $\mu$), using $E_{\mathrm{b}}/E_{\mathrm{F}} = 0.0462$. The system is in the non-topological [topological] superfluid phase for all data points shown in panel (a) [(b)]. The critical Zeeman energy $h_{\mathrm{c}}$ is equal to $0.507\,E_{\mathrm{F}}$ [$0.356\,E_{\mathrm{F}}$] for the situation depicted in panel (c) [(d)].

$2 \times 2$ theory should be more accurate at larger $h$. The situation shown in our Fig. 7(a) corresponds reasonably closely to the case for which $|\Delta|$ *vs.* $\lambda$ is plotted in Figs. 4(a) and 4(b) in Ref. [32] (they have a larger $E_{\mathrm{b}}/E_{\mathrm{F}}$ and smaller $h/E_{\mathrm{F}}$), and there is excellent qualitative agreement between their results and ours.

The exact results for the $|\Delta|$-*vs.*-$h$ dependence given in Fig. 7(c) agree with those available from Refs. [33,34]. For small $h$, deviations between the values calculated within the projected $2 \times 2$ formalism and the exact $4 \times 4$ theory are significant, and even the qualitative behavior exhibited by the respective $|\Delta|$-*vs.*-$h$ dependences is seen to be quite different. However, the agreement becomes quite good in the topological regime realized for $h > h_{\mathrm{c}} = 0.507\,E_{\mathrm{F}}$. In contrast, the projected $2 \times 2$ theory is seen to become overall very accurate, even in the non-topological phase, for the larger value of $\lambda$ for which results are given in Fig. 7(d). Thus, as already indicated by the numbers in Table 1, the effective $2 \times 2$ theory yields quantitatively satisfactory results for sufficiently large values of $2m\lambda/(\hbar^2 k_{\mathrm{F}})$.

# 5 Conclusions and outlook

We have derived an accurate effective theoretical description for superfluidity in 2D Fermi gases with broken spin-rotational invariance due to the presence of spin-orbit coupling and Zeeman spin splitting. Starting from the usually applied self-consistent Bogoliubov-de Gennes (BdG) mean-field theory for $s$-wave pairing in four-dimensional Nambu space [Eq. (1a) with (1b) and (13)], we performed a Feshbach projection onto subspaces associated with fixed spin-$\sigma$ degrees of freedom [Eqs. (4)]. Using also the approximations given in Eqs. (5) that are

informed by inspection of limiting behaviors in the BdG quasiparticle dispersions and ignoring terms of $\mathcal{O}(|\Delta|/h)$, we succeeded in fully decoupling the original $4 \times 4$ BdG equation (1a) into two $2 \times 2$ BdG equations; one for each spin projection [Eq. (6) with (7a)].

Our subsequent investigations focusing on uniform systems at fixed total particle density have demonstrated that the effective two-band descriptions for individual spin subspaces provide a useful theoretical framework for studying the unusual physical properties of this superfluid, including topological effects. In particular, we found that the effective theory faithfully reproduces the relevant physical aspects of the Bogoliubov-quasiparticle dispersion with the chiral-$p$-wave-like gap [see Figs. 1(b) and 2] and the occupation-number distribution in reciprocal space (see Fig. 3). For both the dispersions and reciprocal-space density distributions, the projected-$2 \times 2$-theory's accuracy is excellent in the topological regime but generally very good even within a finite range on the non-topological side of the transition. As the Zeeman spin-splitting energy $h$ decreases, so does the accuracy of the projected $2 \times 2$ approach. This is most apparent in the comparison of the chemical potentials self-consistently obtained within the exact $4 \times 4$ and approximate $2 \times 2$ approaches, respectively, shown in Fig. 4. Based on the observation of Fermi-surface-like features in the reciprocal-space occupation-number distribution [Figs. 3(c) and 3(f)], we derived analytical formulae for the chemical potential [Eqs. (21) and (23)] that agree very well with the exact results (see Fig. 4).

The self-consistency condition for the $s$-wave pair potential within the $2 \times 2$ theory turns out to be given entirely in terms of quantities relating to the spin-$\uparrow$ states [Eq. (13) where $\Upsilon_{\mathbf{k}}$ as defined in (16) is replaced by $\Upsilon_{\mathbf{k}}^{\uparrow}$ from (25a)]. We devise two routines for achieving full self-consistency within the $2 \times 2$-projected theory. One is based on the simultaneous solution of the self-consistency conditions (13) and (15) using $2 \times 2$-theory results as input: $n_{\mathbf{k}\uparrow} \equiv n_{\mathbf{k}\uparrow}^{\uparrow}$, $n_{\mathbf{k}\downarrow} \equiv n_{\mathbf{k}\downarrow}^{\downarrow} + n_{\mathbf{k}\downarrow}^{\uparrow}$, and $\Upsilon_{\mathbf{k}} \equiv \Upsilon_{\mathbf{k}}^{\uparrow}$ with relevant expressions given in Eqs. (17a), (17b), and (25a). The other, simpler routine solves the self-consistency condition (13) using $\Upsilon_{\mathbf{k}} \equiv \Upsilon_{\mathbf{k}}^{\uparrow}$ as given in Eq. (25a) and with the chemical potential approximated by the analytical expressions from Eqs. (21) and (23). For the parameter ranges explored in this work, both routines yield practically indistinguishable results for $|\Delta|$, thus making it possible to adopt the simpler routine for further exploration of the physical ramifications of the $2 \times 2$-projected theory. Overall, the combination of the projected two-band description for spin-$\uparrow$ states with the analytical formulae for the chemical potential is seen to provide a reliable theoretical description of the system, with impressive quantitative agreement achieved in the limit of sufficiently large, but entirely realistic, values of the Zeeman splitting and spin-orbit coupling (see Figs. 6, 7 and Table 1).

The ability to utilize an effective two-band ($2 \times 2$) theory for describing superfluidity in the 2D spin-split Fermi gas opens up the opportunity to explore in greater detail suggested analogies with chiral-$p$-wave pairing [9]. In particular, based on the demonstrated accuracy of the projected $2 \times 2$ approach for the case of uniform systems, we expect this formalism to also be effective for describing situations with non-uniform order-parameter configurations [39–41] or in the presence of disorder [42, 43]. These scenarios are interesting because they offer possibilities to create and manipulate exotic Majorana excitations spatially [12, 30] or temporally [40]. Future work will address in detail the question of applicability of the $2 \times 2$ approach in such instances and, as appropriate, apply it to inform the design of basic building blocks for fault-tolerant quantum information processing [3, 18].

## Acknowledgements

The authors gratefully acknowledge useful discussions with Philip M. R. Brydon and Ana Maria Rey, as well as Kadin Thompson's technical help with numerical calculations.



**Funding information** This work was partially supported by the Marsden Fund of New Zealand (contract no. MAU1604), from government funding managed by the Royal Society Te Apārangi.

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
