# Peer review of "Accurate projective two-band description of topological superfluidity in spin-orbit-coupled Fermi gases"

_SciPost Physics, doi:SciPost Phys. 5, 016 (2018)_

## Round 1 · Referee Report · Anonymous (Referee 1) · 2018-5-1

Strengths

1- New, simplified approach to superconductors with spin-orbit and Zeeman couplings in the topological phase. 2x2 matrix instead of 4x4, several analytic expressions result.

2- Writing and figures are clear.

Weaknesses

1- Motivation for and payoff of the 2x2 formulation could be clearer.

Report

Brand et al.'s paper study attractive fermions with Zeeman + spin-orbit couplings, a well-studied situation due to its importance for topological superfluidity. They eliminate the spin-down degrees of freedom that are far off in energy. The formalism they use is a formally exact Feshbach projection technique, together with an approximation that approximates a certain operator appearing in these expressions by its large-k limit. This reduces finding eigenstates, which normally require diagonalizing 4x4 matrices, to diagonalizing 2x2 matrices. Analyzing this, and occasionally employing other approximations, they arrive at simple analytic expressions to describe the topological superfluid in this system. They show the validity of the approach by comparing the exact and approximate calculations.

The problem of topological superfluidity is important, their results appear technically sound and new, and their approximate expressions are at least somewhat simpler to work with than with the solutions to the 4x4 matrix equations. I am not entirely convinced that this formulation sheds substantial new light on the problem, or that the rather modest simplifications justify introducing (sometimes uncontrolled) approximations. Nevertheless, the techniques should be made available to the community, in case they are useful, and so I believe that with revision this paper is suitable for publication.

I have a few comments and suggestions I request the authors to consider, listed below.

Requested changes

1- Is there a limit where the key decoupling approximation, just below Eq 5, can be shown to be accurate? E.g if Delta/h<<1 or something similar? This issue of where, at least in principle, we can expect the approximation to be valid (independent of numerical verification) should be addressed.

2- Comparisons of the 2x2 and 4x4 theories are shown for a rather limited set of parameter values. It would be valuable if, at least for the types of comparisons shown in Fig 1, comparisons at multiple other values of system parameters were shown. This should include other values where the approximation is accurate, but also where it breaks down, giving an indication of what the region of applicability of the present theory is.

3- How does this approximation compare to doing 2nd order degenerate PT in lambda (for example through a Schrieffer-Wolff transform)? Is it equivalent? Does the present approach capture things that this doesn't?

4- On page 7, the authors refer to "some ambiguity in the prefactor." Is this just due to ambiguities in conventions to define a_2D? If so, the authors should state so. If not, the authors should explain what they're referring to.

5- On page 8, the authors find that the sum in 17b diverges logarithmically at large k. This is a little surprising since the approximation used to simplify the Feshbach-projected equations was exact for large k. What is the reason for this divergence, and why is the remedy used appropriate?

6- Figure 2 dicusses self-consistent vs non-self-consistent results. In the non-self-consistent results, exactly what calculation was done?

7- For the chemical potential, is there insight as to why21-23 are so much more accurate that the other 2x2 theory?

8- Perhaps most essentially, I would like to see stronger arguments and examples for results or insights that are significantly easier to obtain in the 2x2 theory than the 4x4 theory. After all, plotting eigenvalues of a 4x4 matrix as a function of some parameter is extremely easy. Some of the analytic results they have provided are clear examples, but if there are other examples, it would help convince readers of the utility of the approach and help them use it for themselves.

---

## Round 2 · Author Response

We would like to thank the referee for the supportive evaluation of our work and constructive feedback on the manuscript. In the resubmitted version, we have taken all suggestions for improvement onboard. Here we respond in detail to each point raised by the referee. A complete list of changes made in the resubmitted manuscript is provided separately.

To 1: The decoupling approximation made in Eq. (5) relies on neglecting terms of order |Delta|^2/(h_k epsilon_k) and becomes exact when Delta = 0. For finite Delta, the dispersion relations and eigenvectors have the correct asymptotic behavior for large k. Integrated quantities will indeed become correct when |Delta|/h << 1. We have modified the discussion of Eq. (5) to clarify the nature of the approximation. We have further added a paragraph with pertinent comments at the end of section 2 ("By construction ... |Delta|\ll h."). A caveat to the limit |Delta|/h << 1 is that it is not sufficient if the approximate self-consistent Delta obtained in the projected 2x2 approach is small, as this can occur by accident during the summation of Eq. (13), but it is required that the exact value of Delta is small. A relevant comment has been added to the discussion of Fig. 7 in Sec. 4.2 ("From the derivation ... self-consistent |Delta|.").

To 2: We are happy to follow the suggestion to show more extensive comparisons between the 2x2 and 4x4 theories. For this purpose, we have included the new Figure 2 and also revised Figure 3 (former Figure 2) to be more informative about the limits where the 2x2 theory breaks down.

To 3: The perturbative Schrieffer-Wolff (SW) transformation and our 2x2-projection technique are formally different, and the approximate dispersions obtained by these approaches become accurate in opposite limits. Both agree to lowest order in 1/h near the topological gap. The SW result is ill-defined at the s-wave gap, while our 2x2-projection result is well-behaved everywhere. SW approximates the exact dispersions better than our result for small k, but SW differs significantly from the exact dispersions at large k, while our 2x2-projection approach exactly captures the latter. Both the divergence at the s-wave gap and the incorrect large-k behavior make it impossible to use the SW-transformation results as input for solving the self-consistency conditions. We include a brief comment comparing our approach to the SW transformation in the revised manuscript at the end of section 2 ("Our Feshbach-projection ... as small quantities.").

To 4: The referee is correct; it is the ambiguity in the conventions for defining the 2D scattering length that has caused different expressions to be given in the literature for the prefactor in the relation between E_b and a_2D. We have clarified this point in the new footnote 4.

To 5: We are very grateful to the referee for raising this issue. Prompted by this comment, we have re-examined the properties of the contributions (17b) to densities and found that there is no logarithmic divergence as we had mistakenly stated. Rather, we have identified the term n_{k\uparrow}^\downarrow to be pathological due to an unphysical pole caused by the denominator of (17b) having a zero in this case. Otherwise this term contributes negligibly to n_{k\uparrow}, hence we continue to omit it. However, the perfectly well-behaved n_{k\downarrow}^\uparrow is now included when calculating particle densities within the 2x2 approach, significantly improving agreement with the exact 4x4 results. See the revised Figure 3.

To 6: We have revised this figure (current Figure 3, former Figure 2) to only include self-consistent results. Our new choice of fixed parameters in the calculation of densities shown in the figure panels is intended to also address the referee’s suggestion (made in point 2 above) to show more broadly how 2x2 and 4x4-theory results compare.

To 7: The previous discrepancy between values for the chemical potential obtained within the 2x2 theory and the analytical approximations of Eqs. (21) and (23) has largely been rectified by including the contribution n_{k\downarrow}^\uparrow to the total density. The chemical potential is now very well reproduced by the 2x2-projected approach, except at small Zeeman energy h where the Feshbach projection is expected to fail. See Figure 4.

To 8: Our main motivation for developing the projected BdG equations was to simplify the numerical procedures for solving the inhomogeneous and time-dependent BdG equations, which will be the subject of future work. We have added a new paragraph to the introduction (“One of the main ... Eq. (20)].”) in order to comment more explicitly on the benefits of our approach.

---

## Round 2 · List of Changes

Changes are listed in the order in which they appear in the manuscript:

- added paragraph "One of the main benefits ... [see Eq. (2)]." in Sec. 1

- reformulated discussion around Eq. (5)

- added new Figure 2

- added discussion of Figure 2 and two paragraphs at the end of Sec. 2 ("Further comparison between ... |Delta|\ll h."), including also new footnotes 2 and 3

- added new footnote 4

- revised Figure 3 (former Figure 2) and reformulated discussion pertaining to this figure in the paragraph below Eq. (17b)

- revised Figure 4 (former Figure 3) and reformulated discussion pertaining to this figure in the paragraph below Eq. (23)

- revised Table 1 to also show results for Delta^{2x2} and mu^{2x2}, rationalised color scheme, and updated discussion pertaining to the table in paragraphs below Eq. (26)

- moved part of the first paragraph of Sec. 4.2 into new footnote 6 and added new paragraph "From the derivation ... self-consistent Delta."

- reformulated parts of the 2nd and 3rd paragraphs in Sec. 5 to reflect revisions to Figures 3 and 4

- included new Refs. [11,39,49,51]

You are currently on this page

Resubmission 1803.05579v2 on 19 June 2018

---

## Editorial Decision

published